

# Mitigating Radome Induced Bias in X-Band Weather Radar Polarimetric moments using Adaptive DFT Algorithm.

Thiruvengadam PADMANABHAN[1,2], Guillaume LESAGE[1], Ambinintsoa Volatiana RAMANAMAHEFA[1], and Joël VAN BAELEN[1]

[1]LACY, Université de La Réunion/CNRS/MétéoFrance, Saint-Denis, La Réunion.
[2]School of Meteorology, University of Oklahoma, Norman, Oklahoma, USA.

*Correspondence to*: Thiruvengadam PADMANABHAN (thiruvengadam7892@gmail.com) and Joël VAN BAELEN (joel.van-baelen@univ-reunion.fr)

**Abstract.** In recent years, the application of compact and cost-effective deployable X-band polarimetric radars has gained in popularity, particularly in regions with complex terrain. The deployable radars generally use a radome constructed by joining multiple panels using metallic threads to facilitate easy transportation. As a part of the ESPOIRS project, Laboratoire de l'Atmosphère et des Cyclones has acquired an X-band meteorological radar with four panel radome configuration. In this study, we investigated the effect of the radome on the measured polarimetric variables, particularly differential reflectivity and differential phase. Our observations reveal that the metallic threads connecting the radome panels introduce power loss at vertical polarization, leading to a positive bias in the differential reflectivity values. To address the spatial variability bias observed in differential reflectivity and differential phase, we have developed a novel algorithm based on the Discrete Fourier Transform. The algorithm's performance was tested during an intense heavy rainfall event caused by the Batsirai cyclone on Reunion Island. The comparative and joint histogram analysis demonstrates the algorithm's effectiveness in correcting the spatial bias in the polarimetric variables.

**Keywords:** Radome joints, Spatial bias, X-band Weather radar, ZDR, PHIDP, Adaptive DFT algorithm, Polarimetric variables, Batsirai cyclone, Multi-panel radome.

## 1. INTRODUCTION

Doppler weather radars provide comprehensive precipitation measurements at high spatial resolution, offering valuable insights into the orography-influenced precipitation processes (Georgis et al., 2000). Among different frequency bands, X-band radars have gained popularity due to their compact size and affordable cost, making them well-suited for deployment in areas with challenging topography. Additionally, modern polarimetric technology has dramatically enhanced the ability of X-band radars to estimate precipitation, overcoming their traditional limitation of high attenuation rates in the rain (Ryzhkov et al., 1994). The ESPOIRS (Etude des Systèmes Précipitants de l'Océan Indien par Radars et Satellites) project implemented by LACy (Laboratoire de l'Atmosphère et des Cyclones) has utilized the advantages of X-band polarimetric radar to examine the dynamics and variability of intense tropical precipitation at the local scale, particularly its interaction with the relief of the island. Initially, an X-band radar operating at 9.41 GHz was deployed south of Reunion Island (Figure 1a) to investigate heavy rainfall and the pronounced relief effects that may contribute to rainfall intensification. Later, the radar was installed in Seychelles and Madagascar to explore different geographical and meteorological contexts.



Although X-band polarimetric radar offers advantages over single polarization radars, maintaining the accuracy of its polarimetric data requires rigorous quality control (Park et al., 2005). One factor that can impact the performance of the radars is the radome. While they play a protective role, radome can passively affect the observations due to material-related transmission losses and variations in sidelobe levels (Figueras i Ventura et al., 2021; Mishra et al., 2014). Mainly, radome panels with metallic threads at their joints, common in mobile radars, can introduce biases in polarimetric observations (Figueras i Ventura et al., 2021; Gourley et al., 2006). The ESPOIRS X-band radar is protected by a composite radome with a 1.435m base diameter and 2m height, consisting of four fiberglass-reinforced lateral segments attached through horizontally aligned metal nuts and bolts arranged along the vertical axis (Figure 1b, 1c). Radar data from the Batsirai cyclone on 3 February2022 show significant biases in differential reflectivity (ZDR) and differential phase (PHIDP). One of the objective of this study is to identify the source of these biases and hypothesize that they originate from the radome metal joints.

In similar scenarios, efforts have been made to address biases using spatial calibration curves from radar data (Figueras i Ventura et al., 2021; Gourley et al., 2006). However, the stochastic nature of radar measurements introduces inherent uncertainty into calibration curves, leading to an overall increase in measurement uncertainty. Previous investigations have also recommended the implementation of a seamless monoblock radome to mitigate spatial bias, as it eliminates the presence of joints and, subsequently, any bias. Nevertheless, the unique design of the portable ESPOIRS X-band radar, which prioritizes ease of transportation, presents inherent challenges in adopting a monoblock radome solution. To address this challenge, this study proposes an adaptive Discrete Fourier Transform (DFT) based algorithm to mitigate the bias. This algorithm effectively utilizes the information available in radar scans and enables the correction of the ZDR and PHIDP moments for various scan strategies.

The distinguishing feature of this algorithm is its capacity for dynamic threshold adjustment in the correction of frequency values generated by the DFT based on the characteristics of radar moments. Such adaptivity sets this algorithm apart from traditional methods (Figueras i Ventura et al., 2021; Gourley et al., 2006), which employ fixed spatial calibration curves. To assess the performance of the proposed algorithm, it is applied to correct the ZDR and PHIDP radar scans obtained during an intense heavy rainfall event caused by the passage of the Batsirai cyclone in the vicinity of Reunion Island on February 2 to 4, 2022. Section II focuses on the source of the spatial bias in ZDR, offering insights into the factor responsible for this bias. The adaptive DFT algorithm proposed in this study is introduced in section III, providing a discussion of its mathematical formulation. Section IV assesses the proposed algorithm's performance through comparative and joint histogram analysis. Section V summarizes the key points of the study and outlines the direction of future research to test further and enhance the algorithm's performance.

## 2. SOURCE OF BIAS

To elucidate the origins of positive bias in PHIDP measurements and ascertain its underlying causes, this study employs a comprehensive approach involving the calculation of PHIDP offsets for all PPI scans conducted on the 2/3/2022 case study, following the technique outlined in previous works (Figueras i Ventura et al., 2012), which utilized PHIDP offsets to discern the influence of radome joints. The findings in Figure 1d exhibit a box plot distribution of PHIDP offsets across various azimuth angles. Figure 1d shows a sinusoidal azimuthal pattern with four periods in the PHIDP offset. Strikingly, a distinct correlation emerges between the azimuthal period positions of the PHIDP offsets and the orientation of the radome joints, with notable alignment occurring at approximately 355, 85, 175 and 265 degrees. This observation strongly suggests that the metallic threads and joint configuration are the key factors contributing to the bias in PHIDP. The localization of the bias near the joints is due to signal phase sensitivity to path length and material properties. Radome metal joints can introduce significant phase shifts in specific



polarizations, impacting PHIDP measurements when signal paths intersect with these joints (Li et al., 2017), explaining the
observed PHIDP offset collocation with the radome joint direction.
The positive bias observed in ZDR (Figure 2a) can result from either an increase in horizontal reflectivity values (ZH) or a decrease
in vertical reflectivity values (ZV). To ascertain the cause of the positive bias in ZDR, the temporal median of horizontal and
vertical reflectivity obtained from RHI scan angles of 28, 300, 45 and 76 degrees on 3rd February 2022 were computed.
Subsequently, for each median RHI scan at 28, 300, 45 and 76 degrees of azimuth respectively, elevation angles exhibiting positive
ZDR bias (70, 70, 36, and 34) and unbiased elevation angles (66, 67, 31, and 27) were identified for both horizontal and vertical
polarization and compared as shown in Figure 2a-2d. The biased (red, blue) and unbiased (black, green) elevation angles were
chosen sufficiently close to each other to ensure that the same hydrometeor characteristics were observed.
Figures 2a and 2b show that the horizontal reflectivity values in biased and unbiased elevation angles (red, black) exhibit a
consistent covariation pattern with a negligible offset. Consequently, one would expect the vertical reflectivity values (blue, green)
to follow a similar trend. However, it is observed that the vertical reflectivity values correlate with each other but with an offset.
Specifically, the biased elevation angle's vertical reflectivity (blue) has a negative offset compared to the unbiased elevation angle's
vertical reflectivity (green). At lower elevation angles of RHI scans, where the variance of reflectivity values across the range
direction is limited, a noticeable decrease in vertical reflectivity in the biased region is prominent. Specifically, at lower elevation
angles of the RHI scans at 45 and 76 degrees of azimuth (Figure 2c and 2d), the vertical reflectivity values in the biased region are
noticeably lower than their corresponding horizontal and vertical reflectivity values in the unbiased region. This indicates that the
metal joints have exerted a notable influence on the propagation of microwaves, particularly regarding vertical polarization. This
influence has led to a systematic bias in ZDR.
From Figures 4a and 4c, it can be noted that PHIDP and ZDR exhibit biases at somewhat different azimuth angle locations. The
observed bias in ZDR at varying angles, compared to the bias in PHIDP, can be attributed to distinct scattering characteristics. The
ZDR's azimuthal bias could arise from factors related to the scattering of incident power from the main lobe around the metallic
threads, which could potentially amplify the impact of the side lobes. These increased side lobe levels cause more energy to be
radiated or received in undesirable directions, potentially amplifying susceptibility to noise or interference from those directions
(Frech et al., 2013). This differs from PHIDP's bias, which is primarily influenced by the path length the radar signals traverse (Li
et al., 2017). Hence, the PHIDP bias are observed in the direction of the radome joints. Detailed simulations or measurements are
imperative for a comprehensive understanding of the precise impact of radome joints on side lobe levels, phase delays, and their
correlation with ZDR and PHIDP.
**3.      ADAPTIVE DFT ALGORITHM**
This study introduces a DFT algorithm to mitigate the spatial bias present in the polarimetric variables. The correction procedure
starts by performing a Discrete Fourier transform on a single radar scan, which, in this case, corresponds to the 76° azimuth RHI
scan measured on 02/03/2022 01:08 UTC (Figure 3a). A one-dimensional DFT is applied along the range direction to process the
polarimetric variable x for a particular azimuth or elevation angle θ and N total range gates, as shown in equation 1, where n and
k represent the range gate and frequency index, respectively. The transformation F[k] is repeated for each angle θ, respectively
azimuth for PPI scans and elevation for RHI scans, systematically covering the entire radar scan to obtain F[θ, k].





Subsequently, the power spectrum ($P[\theta, k]$ ) is calculated from the obtained Fourier transform ($F[\theta, k]$) for each azimuth/elevation
angle, as depicted in Figure 3c. In Figure 3c, a three-dimensional representation is shown, illustrating the power spectrum obtained
from the DFT for each elevation angle of the RHI scan. Notably, the power value of the zero frequency component ($P[\theta, 0]$) derived
from various elevation angles, representing the DFT's DC component, reveals three distinct peaks corresponding to the spatial bias
pattern observed in Figure 3a's RHI scan. This indicates that a bias caused by the radome joints will result in a constant positive
increase of the magnitude all along the range, an offset, which will show up as an increase in the power value of the DC component
of the DFT spectrum.

$$F[\theta, k] = \sum_{n=0}^{N-1} e^{-2\pi j \frac{kn}{N}} x[\theta, n] \quad \text{for} \ \ 0 \le k < N \qquad (1)$$


To address this bias, a correction factor (A/B) is introduced to filter the three peaks (Figure 3d) associated with the spatial bias.
Notably, the correction factor is only applied to the DC component values ($F[\theta, 0]$) that exceed a certain adaptive threshold.

$$F[\theta, 0]' = \begin{cases} F[\theta, 0] \times \frac{A}{B} & \text{if } P[\theta, 0] > \text{Median}(P[\theta, 0]) \\ F[\theta, 0] & \text{otherwise} \end{cases} \qquad (2)$$


The threshold value is determined based on the moving median window of the power spectrum at the zeroth frequency
($\text{Median}(P[\theta, 0])$). The correction factor is calculated as the ratio between the median values of $F[\theta, 0]$ corresponding to the
power values below the threshold (A) and the median values of $F[\theta, 0]$ corresponding to the power values above the threshold
(B).

$$A = \text{Median}(F[\theta, 0]) | P[\theta, 0] < \text{Median}(P[\theta, 0]) \qquad (3)$$

$$B = \text{Median}(F[\theta, 0]) | P[\theta, 0] > \text{Median}(P[\theta, 0]) \qquad (4)$$

$$P[\theta, 0] = \left[ \sqrt{\text{Real}(F[\theta, 0])^2 + \text{Img}(F[\theta, 0])^2} \right]^2 \qquad (5)$$

Consequently, the zeroth frequency values that surpass the threshold are multiplied by the calculated correction factor, while the
remaining values are left uncorrected. Figure 3d shows the power $P[\theta, 0]$ from the DC component is characterized by distinct peaks
alongside the depiction of the adaptive threshold values and the reduced power after applying the correction factor. To derive the
corrected ZDR image (Figure 3b), an inverse Fourier transform is performed on all frequencies, encompassing the corrected DC
component.

$$x[\theta, n] = \frac{1}{N} \sum_{k=0}^{N-1} e^{-2\pi j \frac{kn}{N}} F[\theta, k] \qquad (6)$$


**4.      ASSESSMENT OF ADAPTIVE DFT FILTERING ALGORITHM EFFICIENCY**
The performance of the adaptive DFT filtering algorithm is assessed across different scan strategies (PPI and RHI) and different
polarimetric moments (ZDR and PHIDP) to determine its effectiveness. Figures 3a and 4a show increased spatial variability of
ZDR at specific azimuth and elevation angles of RHI and PPI scans, respectively, with a maximum variability of around 1.5-2 dB
observed. Similarly, Figures 4c and 5a depict increased spatial variability of PHIDP in PPI and RHI scans, with a magnitude of



variability around 8 to 10 degrees. The application of the algorithm led to a substantial reduction in variability, as shown in Figures
3b, 4b, 4d and 5b. The proposed algorithm reduced the spurious ZDR variability from 1.5-2 dB to 1-1.34 dB, and the PHIDP
variability was reduced to less than 5 degrees. The results indicate the algorithm's effectiveness in mitigating spatial variability in
both ZDR and PHIDP values across all elevation angles and scanning strategies.
The relationship between ZDR and ZH provides insights into how raindrops deviate from a perfect spherical shape. Larger
raindrops exhibit higher horizontal reflectivity and have their long axis oriented in the horizontal direction compared to the vertical
direction, thereby showing signatures of higher ZDR values. Conversely, for lower values of horizontal reflectivity (less than 30
dBZ), we generally expect a ZDR value of nearly zero dB due to the near spherical shape of small drops. Therefore, comparing
the relationship between ZH and ZDR before and after applying the algorithm (Figure 6) shows how the correction has brought
the biased ZDR close to the expected value. Figures 6a and 6b show the joint histogram of ZDR and ZH for all the PPI scans
conducted on 3 February 2022 before and after applying the correction algorithm, with the marginal distribution of ZH in the top
and the marginal distribution of ZDR on the right side. Likewise, figures 6c and 6d show the joint histogram of ZDR and ZH for
all the RHI scans on the same date, before and after applying the correction algorithm. As these figures capture the data before any
attenuation correction, the reflectivity distribution ranges from light drizzle (near zero dBZ) to heavy rainfall ($\cong$40 dBZ). It can be
observed from Figure 6a, 6c that the distribution of ZDR extends significantly beyond 2dB for reflectivity values below 30 dBZ.
Such increases arise from the spatial positive bias in ZDR discussed in view of Figures 3a and 4a. The impact of the developed
algorithm on the distribution of ZDR is highlighted in Figures 6b and 6d. Following the correction, the mean distribution of ZDR
shifts towards values closer to zero, approaching the theoretical values at lower reflectivity. Furthermore, the correction also results
in a noticeable increase in negative values of ZDR, indicating the presence of differential attenuation fingerprints, a common
occurrence in X-band radar systems, which were not as discernible before the correction. Figures 6b and 6d reveal the impact of
the algorithm in successfully addressing the positive bias in ZDR.
## 5.    CONCLUSION
This study investigates the impact of radome joints on the polarimetric variables of a portable X-band weather radar. The radome
design introduces spatially dependent biases in polarimetric variables due to preferential loss of power in vertical polarization
caused by metallic thread alignment. Due to the stochastic nature of radar measurements, these spatial errors exhibit variability
across different scans. This study proposes a novel adaptive DFT algorithm to address the changing bias. The algorithm is
formulated with the understanding that the bias caused by the radome joints remains constant along the range. Consequently, the
DFT applied along the range exhibits an increase in magnitude in the zeroth frequency component (DC) for azimuth angles aligned
with the radome joint positions compared to those misaligned. This leads to varying offset values of polarimetric variables across
azimuths and elevations depending on the radar beam position with respect to radome joint positions. The algorithm detects and
reduces the spatial bias pattern by suppressing the increase in offset values.
The algorithm's performance in mitigating the positive spatial bias in ZDR and PHIDP is evaluated by applying the algorithm in
the PPI and RHI polarimetric variable scans measured during the Batsirai cyclone event on 3 February 2022. The results show that
the adaptive DFT algorithm presented herein effectively addresses the spatial biases, enhancing the accuracy of ZDR and PHIDP
measurements regardless of the scan type.
As the primary source of spatial bias originates from vertical polarization, the algorithm's effectiveness is further validated by
examining the relationship between ZH and ZDR values. Notably, the correction reduces the positive bias observed in ZDR at



lower reflectivity levels, bringing the mean ZDR value closer to the expected levels. Additionally, the algorithm reveals negative
ZDR values, indicating the presence of attenuation fingerprints.
The proposed algorithm has limitations related to sample size and precipitation type. To effectively distinguish between
meteorological targets and bias, it requires a large, noise-free sample along the range direction. Additionally, its performance relies
on suppressing variations in the magnitude of the DC component of DFT observed across continuous azimuth angles. As a result,
the algorithm performs optimally in scenarios characterized by uniform precipitation, such as stratiform precipitation and large
convective systems that cover extensive areas around the radar. However, for isolated or scattered precipitation systems located in
limited range and azimuth gates, the algorithm's performance needs to improve due to its inability to capture variations in the DC
component magnitude derived from the DFT. To advance future research and potential improvements, an area of interest lies in
evaluating the proposed algorithm's performance by conducting measurements both with and without the radome. An additional
consideration is the possibility of incorporating a seamless monoblock radome as a future plan.

**Data availability**
All radar data is available at https://geosur.osureunion.fr/thredds/catalog/researchprogram/espoirs/1-
Saint_Joseph/RADAR/Data/az-vol-75-0125-1to25deg-2022-01/2022/02/catalog.html. For information on rain gauge data, please
send requests to ambinintsoa.ramanamahefa@univ-reunion.fr or visit the Météo France website https://portail-
api.meteofrance.fr/web/fr/api/DonneesPubliquesObservation
**Code avalaibility**
The code is available upon request from the authors
**Author Contributions**
Thiruvengadam PADMANABHAN developed the algorithm, implemented the data analysis and wrote the paper, Guillaume
LESAGE and Ambinintsoa Volatiana RAMANAMAHEFA contributed to the implementation of the methodology and revised the
paper, Joël VAN BAELEN revised the paper and led the ESPOIRS project and radar field deployments.
**Competing interests**
The authors declare that they have no conflict of interest.
**Acknowledgment**
This work is part of the INTERREG V ESPOIRS project (Study of Precipitating Systems in the Indian Ocean by Radar and
Satellites). The ESPOIRS scientific program is led by LACy (University of La Réunion / CNRS / MétéoFrance) and funded by the
European Union (FEDER program - GURDTI/20201589-0021087), the Réunion Region, SGAR-Réunion, the French State (
CPER) and the University of La Réunion.



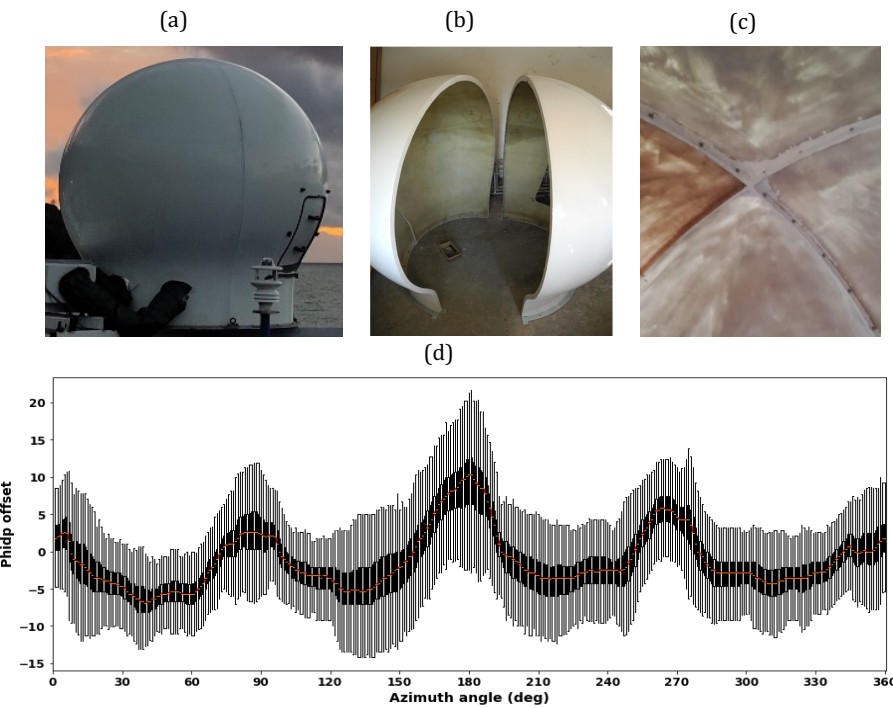

**Fig. 1.** Photographs illustrating the structure and assembly of the four lateral panels of the ESPOIRS Radar radome showing the regions of metal nuts and bolts (a), (b), (c). Distribution of PHIDP offset obtained from all the PPI scans observed on 3rd February 2022 (d).



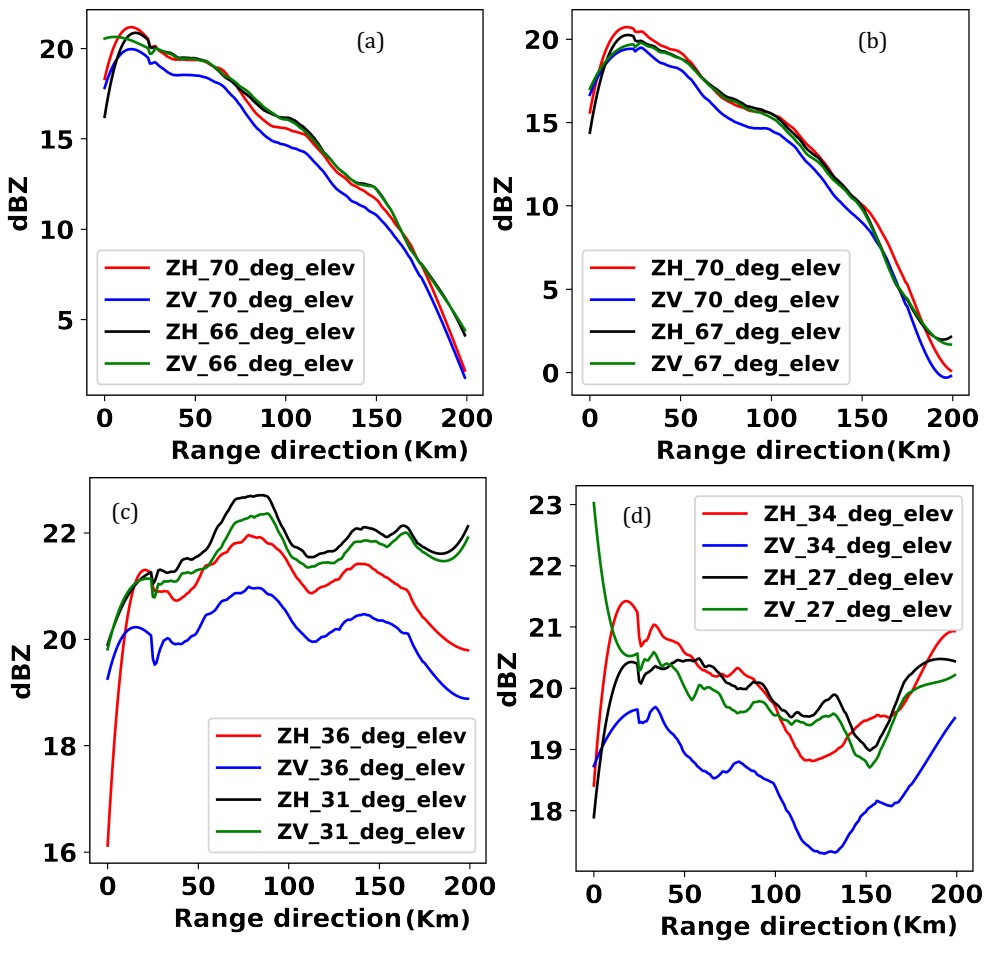

**Fig. 2.** The temporal median of horizontal and vertical reflectivity was obtained from RHI scan angles of (a) 28°, (b) 300°, (c) 45°, and (d) 76°, with corresponding elevation angles showing positive ZDR bias in red and blue (70°, 36°, and 34°) and unbiased elevation angles in green and black (66°, 67°, 31°, and 27°) on February 3rd, 2022.

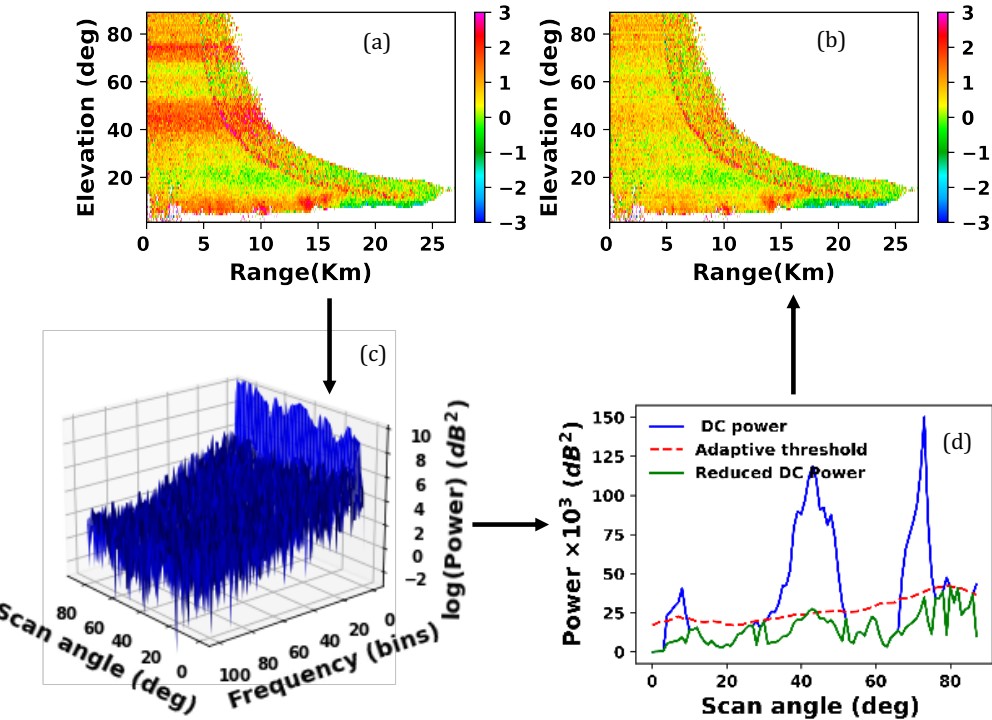

**Fig. 3**.  76° RHI scan of ZDR measured on 3rd February 2022 01:08 UTC before and after correction (a,b), 3D power spectrogram (c), The
power from zeroth frequency (DC) component (blue), adaptive threshold values (dotted red) and the reduced zeroth frequency power after the
application of the correction factor (d).



**Fig. 4.** 29° PPI scan of ZDR, PHIDP before correction (a, c) and after correction (b, d), horizontal (e) and vertical reflectivity (f) measured on 03 February 2022 01:50 UTC.





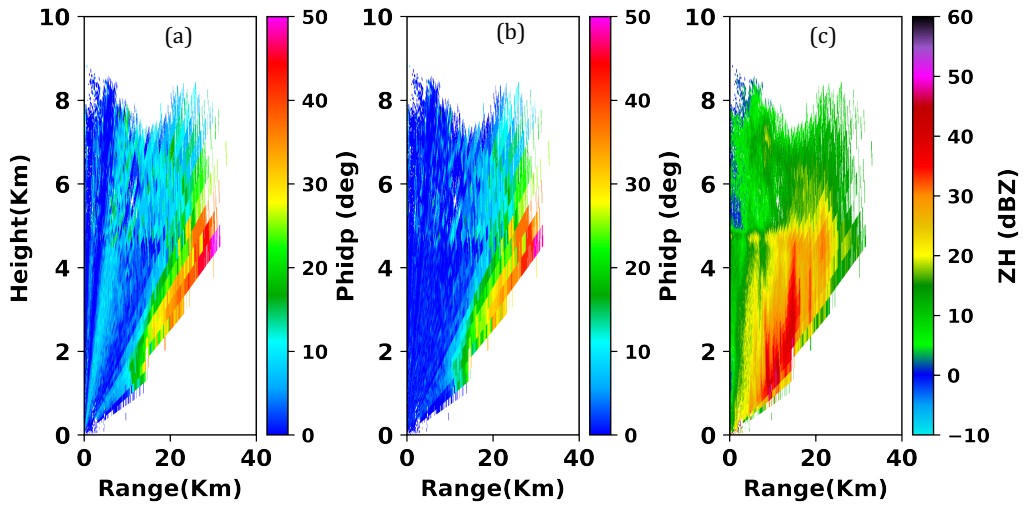

**Fig. 5.** 28° RHI scan of PHIDP before and after correction (a, b) along with ZH (c) on 03 February 2022 01:28 UTC.

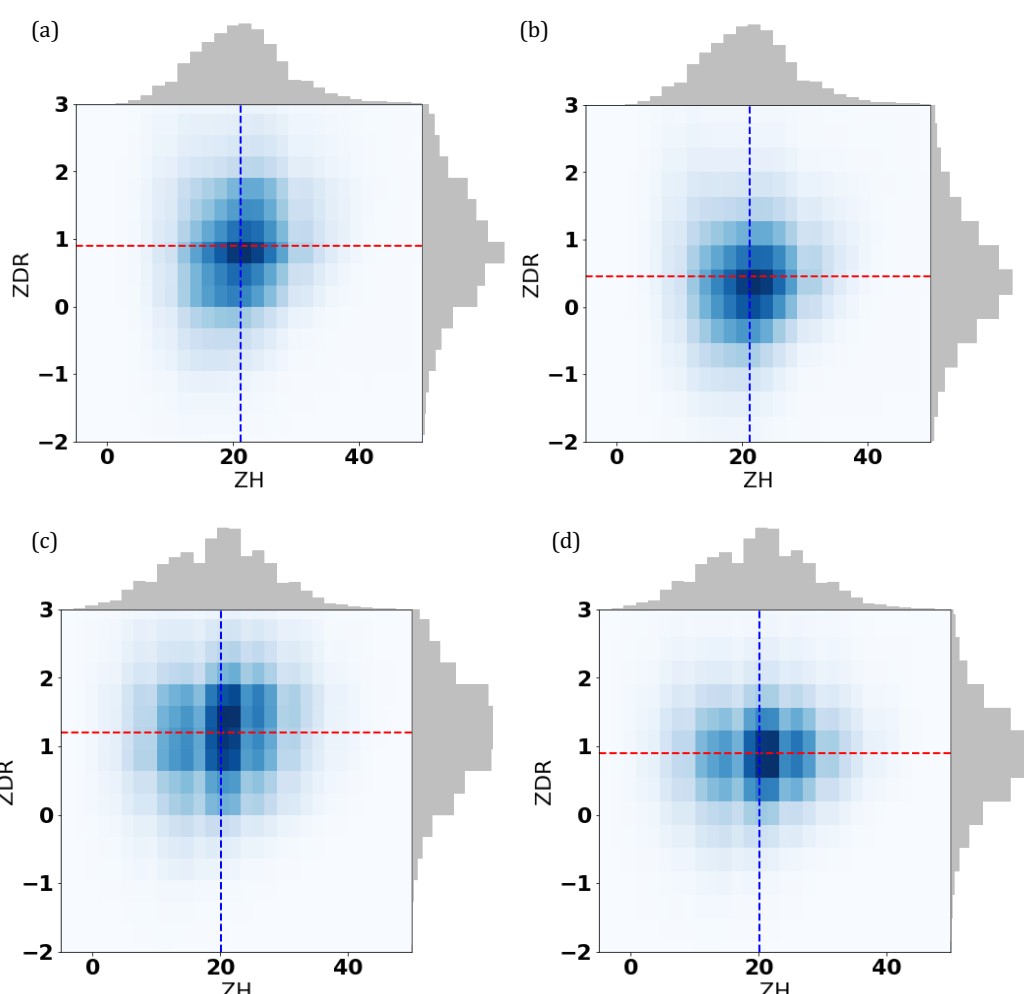






**Fig. 6.** Joint histograms of ZDR and ZH for (a) PPI scans before and (b) after correction, as well as (c) RHI scans before and (d) after correction, all from scans conducted on February 3, 2002, with the marginal distribution of ZH in the top and the marginal distribution of ZDR on the right side. The red dotted line represents the median of ZDR, while the blue dotted line represents the median of ZH.

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
