# Peer review of "Mitigating Radome Induced Bias in X-Band Weather Radar"

_Atmospheric Measurement Techniques, 2024_

## Author Response (AR1)

**Response to Reviewers' Comments**

**Reviewer 1:**

**Comment 1:**

**Comment 1.1:**

With the same method, the author tried to correct the "azimuth" variability of ZDR and PHIDP on PPI scan and to correct "vertical" variability on a RHI scan.

Figure 1 Phidp on **PPI** scans; Figure 2 Some ZDR profiles on **RHI** scans in a sub-figure; Figure 3 Illustration of the method based on a **RHI** scan of ZDR; Figure 4 Application of the method for PHIDP and ZDR on **PPI** scans; Figure 5 Application of the method for PHIDP on **RHI** scan.

According to my knowledge, it is too ambiguous to mix all these contents in the same investigation and attribute all of them to the radar radome.

For the RHI scan, it is well known that ZDR depends on the elevation angle (see Bechini et al. 2008: Differential Reflectivity Calibration for Operational Radars). If elevation angle is equal to 90°, the ZDR is expected to be around 0 dB, while ZDR is much larger than 0 dB for the elevation angle equal to 0°. This is totally due to the geometry of raindrops shape, but nothing related to the radome.

**Response 1.1:**

Thank you for your insightful comments and for referencing the Bechini et al. (2008) study. You are correct that ZDR depends on the elevation angle due to the geometry of raindrop shapes:- the ZDR decreases with increasing elevation angle. However, our study detected an anomalous increase in ZDR at certain elevation angles in our RHI scans, which suggests the presence of bias at specific elevation angles. We believe these biases are introduced by the radome joints, affecting the signal differently at various elevation angles.

Our correction algorithm is specifically designed to address these bias at specific elevation and azimuth angles. It does not alter the expected property of ZDR variation due to raindrop geometry. In Figure A (attached), we present the average ZDR observed across different

elevation angles in stratiform rainfall from the  $300^{\circ}$  RHI scan measurements between the 1 km and 2 km range near the radar over 10 consecutive time steps, focusing on uniform precipitation below the bright band. The  $300^{\circ}$  RHI scan angle was chosen because it exhibits minimal ground clutter at close range, due to the absence of significant orography in this direction. Prior to the correction, the ZDR profile showed an unexpected increase at certain elevation angles, deviating from the theoretical decrease expected from raindrop shape considerations alone. After applying our correction algorithm, these anomalies are effectively removed, resulting in a ZDR profile that aligns with the theoretical expectations Fig A(c) (i.e., decreasing consistently with increasing elevation angles). This correction enhances the accuracy of ZDR measurements, thereby fulfilling the criteria necessary to apply the Bechini et al. (2008) algorithm.

Regarding the potential ambiguity in mixing azimuthal variability on PPI scans and vertical variability on RHI scans, our intention was to comprehensively investigate the radome joints' specific impact on our radar measurements in both horizontal and vertical scanning modes.

Figure A : (a) ZDR before applying the algorithm (b) ZDR after applying the algorithm. The blue line indicates the average ZDR (300° RHI) between 1 km and 2 km range near the radar over 10 consecutive time steps. The red line smooths the blue to show the underlying trend.(c) theoretical variation of ZDR from Fig 1 of Bechini et al. 2008.

**Comment 1.2:**

Secondly, to my knowledge, the ZDR (or Phidp) difference at different elevations can be also related to the rotary joints of antenna for some radars.

**Response 1.2:**

Thank you for raising the issue of potential biases in ZDR measurements due to rotary joints, for RHI scans. It is important to note that the radar used for this study is a GAMIC GMWR-25-DP, which by design has the transceiver unit fixed just behind the antenna so that the transceiver and the antenna move together. Therefore, as we confirmed with the manufacturer, only the digital signals at the output of the Intermediate Frequency Digitizer are sent over the optical fiber. This system architecture ensures that there is no degradation of the signal due to the rotary joint. To clarify, the following statement has been appended to the revised manuscript in lines 36-38. "The radar used for this study is a GAMIC GMWR-25-DP, which by design has the transceiver unit fixed just behind the antenna move together and avoid the use of rotary joints."

**Comment 1.3:**

For the PPI scan, the impact of radome is easier to understand (from the figure 1 of the paper). However, when the radome is wet, in the past many studies showed the ZDR bias on PPI scan depending on the direction of wind.

**Response 1.3:**

Thank you for highlighting the concern regarding the potential influence of wind direction on ZDR bias, particularly under conditions where radome wetness could impact the results. In our study, we had the opportunity to observe ZDR bias under dynamic meteorological conditions during the passage of the Batsirai cyclone. During this event, wind directions varied dramatically, shifting from southeast to northwest. Despite these substantial changes in wind direction, our data showed that the direction of ZDR bias remained constant throughout the cyclone's passage. Importantly, this consistent bias was closely aligned with the structural features of the radome, specifically the joints, rather than varying with the wind direction. This

observation provides evidence that the ZDR bias observed in our study was predominantly due to the radome joint itself and not by external meteorological factors like wind direction.

**Comment 1.4:**

So the physical explications of the ZDR and Phidp biases can be very different for the PPI and RHI scans. In my opinions, it will be better to deal with them separately in different sessions.

**Response 1.4:**

Thank you for suggesting the possibility of handling the explications of ZDR and Phidp biases separately for PPI and RHI scans based on potential differences in their sources. In the course of our investigation, we carefully examined the sources of biases in both PPI and RHI scans. Our current findings indicate that the biases observed in both types of scans are predominantly associated with the radome joints. This consistent source of bias across different scanning modes suggests that while the impact of these biases on ZDR and Phidp measurements might manifest differently due to the scanning geometry, their origin remains the same. Given this understanding, we believe that maintaining the discussion of these biases within the same section of our study is appropriate. This approach facilitates a clearer understanding of the underlying issues and the effectiveness of our proposed corrections.

Nevertheless, we acknowledge that further research could reveal more nuanced distinctions or additional sources of biases in ZDR and Phidp readings across PPI and RHI scans. We are open to revisiting this decision in future iterations of our research as more data becomes available.

**Comment 2:**

The author used a moving window on the DC components of the DFT to filter the bias. A Dc component (k=0) is simply the averaged all values of ZDR (or PHIDP) along a radar ray (I called it mean\_ZDR\_ray or mean\_PHIDP\_ray). So the main idea of the paper is to assume the median of mean\_ZDR\_ray (or mean\_PHIDP\_ray) should not be biased. Hence we can use these medians for corrections. The author should carefully deal with the following questions in details:

**Comment - 2.1** How the proposed method can correct an absolute bias? It seems to me that the method can only reduce the spatial variability in ZDR and PHIDP. If there is a global and

constant bias (e.g. 1 dB) induced by radome at all directions (all azimuth directions and all elevations), how the median of a moving window can determine and then filter this constant bias?

**Response :** Thank you for your thoughtful comment.

The proposed method is designed to correct spatial biases in ZDR and PHIDP measurements, not absolute biases that are constant across all azimuth directions and elevations. Correcting absolute biases that are uniform across all measurements was not the intent of our study. Addressing such biases typically requires additional calibration procedures.

We have appended the sentence "The proposed method is designed to correct spatial biases in ZDR and PHIDP measurements, not absolute biases that are constant across all azimuth directions and elevations" on page 2, lines 61 to 63, to clarify that our method is intended for correcting spatially varying biases and does not address absolute biases that are consistent throughout the entire dataset.

**Comment 2.2** The averaged ZDR (or PHIDP) values along a radar ray are impacted by the attenuation as well, particularly for a X-band radar during a tropical cyclone event. We can get very negative ZDR and high PHIDP on some radars rays due to the attenuation. And we see often very large ZDR in the bright band. And these values can have large impact on your median of the moving window. The conventional method takes only "first and short" rain segment (close to radar) to calculate the averaged ZDR and PHIDP offset to avoid the impact of attenuation. Please explain the advantages of your method compared to the conventional method.

**Response:** Thank you for your insightful comment regarding the impact of attenuation and bright band effects on the averaged ZDR and PHIDP values. Our study adopts the conventional practice of using the first 100 rain range gates to calculate the FFT so that we mitigate the impact of attenuation and bright band effects. We apologize for not clearly mentioning it in the document. Now, we have included in the revised manuscript in Page 4 lines 113-114 as "A one-dimensional DFT is applied along the range direction to the first 100 radar gates containing rain near the radar in order to process the polarimetric variable x for a particular azimuth or elevation angle". While our study follows the conventional practice of limiting the analysis to

these gates to mitigate attenuation effects, we acknowledge that in cases of intense rainfall close to the radar, attenuation may still influence.

**Comment 3:** Line 222: In the figure 2, the author illustrates some RHI observed from the X-Band radar. The horizontal axis is marked as range direction in km. I am surprised that the author found the reflectivity equal to 20 dBZ (or 22 dBZ) in the figure 2c at a range of 200 km with elevation angle equal to  $31^{\circ}$  (or  $36^{\circ}$ ). The height of the radar beam at 200 km is equal o 200 km x tan ( $31^{\circ}$ ) = 120 km! 20 dBZ at 120 km of the atmosphere is not impossible for X-band radar to my knowledge.

**Response 3:** Thank you for bringing this to our attention. We apologize for the confusion caused by a labelling error in Figure 2. The horizontal axis was incorrectly labelled as "Range Direction (km)" when it was actually in range gate count, not kilometres. This mislabelling led to a misunderstanding regarding the range and corresponding heights. We have updated Figure 2 to correctly label the horizontal ticks labels in kilometres instead of range gate units. The corrected Figure 2 now accurately represents the range in kilometres, ensuring that the reflectivity values correspond to realistic altitudes for X-band radar observations.

**Comment 4:** - Line 77: The author showed the bias induced by radome on PPI occurring at approximately 355, 85, 175 and 265 degrees (Line 71). Later, the author selected the RHIs at 28, 300, 45 and 76 degrees of azimuth and showed the illustrations in Figure 2. Please justify the selection of these RHIs at 28, 300, 45 and 76 degrees of azimuth? Why the author don't show a RHI at 355 degrees (or at 85, or at 175, or at 265 degrees) of azimuth which is impacted by radome according to the Figure 1?

**Response 4:** Thank you for your comment. We selected the RHIs at azimuths of 28°, 45°, 76°, and 300° because our primary objective was to study orographic precipitation during the cyclone's passage. These azimuths point directly toward mountainous regions where orographic effects were expected to enhance precipitation. We did not observe RHIs at azimuths of 355°, 85°, 175°, and 265°, which were impacted by radome-induced biases, because these directions do not align with our initial areas of interest. We have appended the sentence, "These specific azimuth angles were selected as they were the only available RHI scans for analysis." in page 3 lines 83-84 to clarify our selection.

**Comment 5:** - Line 124 "The threshold value is determined based on the moving median window of " The author should explain the width of this moving median window (how many azimuths/elevations points are used to calculate the median). This is a fundamental parameter of the method. If it is possible, I hope to see why the author selected such width?

**Response 5:** Thank you for highlighting the importance of explaining the width of the moving median window in the proposed method. The threshold value in the proposed method is determined using a moving median window that includes all azimuth angles in a PPI scan, covering a full 360 degrees. For RHI scans, the window encompasses all available 88 elevation angles. This approach also simplifies the implementation, as it removes the need to define specific window sizes that could vary with different scans or atmospheric conditions. By using the full range of azimuths, we apply bias detection and correction uniformly across the entire scan, which maintains consistency in our results. Now, we have included the information on the width of the moving window in the revised manuscript in Page 4 lines 138-139 as "For PPI scans, the moving median window encompasses all azimuth angles, covering a full 360 degrees. In the case of RHI scans, the window includes all available 88 elevation angles. This approach simplifies implementation by eliminating the need to define specific window sizes, which could vary with different scans or atmospheric conditions.

**Comment 6:** - Line 125 : Missing a space between the "of" and " $F[\theta, 0]$ "

**Response 6:** The suggested correction to insert a space between "of" and " $F[\theta, 0]$ " has been implemented in Page 4 line 142.

**Comment 7:** - Line 133 : "the zeroth frequency values that surpass the threshold are multiplied by the calculated correction factor, while the remaining values are left uncorrected." Please express "the remaining values are left uncorrected" by mathematics expressions. If I understand well, it should be F '  $[\theta, k] = F [\theta, k]$  if k <> 0

**Response 7:** Thank you for your comment. We have updated the manuscript to include a clear mathematical expression (equation 3) specifying that for all frequency components  $k\neq 0$ , the values remain unchanged:

$$\mathbf{F}[\theta, \mathbf{k}]' = \mathbf{F}[\theta, \mathbf{k}] \quad for \ k \neq 0 \tag{3}$$

**Comment 8:** Line 138 : In the equation (6), I think the  $F[\theta, k]$  should be the power spectrum after the filter of the correction factor (A,B). But in the equation (1), the author uses the same  $F[\theta, k]$  to represent the power spectrum before the filter. Same  $F[\theta, k]$  in the (6) and (1) leads to confusion.

**Response 8:** Thank you for pointing out the potential confusion regarding the use of  $F[\theta,k]$ . We have modified the equation for the corrected Fourier transform component  $F'[\theta,k]$  and the output  $x[\theta,k]'$  as

$$x[\theta, k]' = \frac{1}{N} \sum_{k=0}^{N-1} e^{-2\pi j \frac{kn}{N}} F[\theta, k]'$$
(7)

**Comment 9:** - Line 239. In Figure 6, the author forgets the color palette to indicate the Count or Frequency of the joint histograms of ZDR-ZH.

**Response 9:** Thank you for pointing out the missing color palette in Figure 6; we have now included it.

**Reviewer 2:**

This article highlights the effects of the radome (in particular its metallic junctions) of an Xband weather radar on the measured values of the ZDR and PHIDP variables. Higher PHIDP values are observed along the axis of the junctions of the radome pieces, and lower Zv values are observed more diffusely, which positively biases the ZDR values. After highlighting these biases and their causes, the authors proposed a correction method based on a DFT algorithm. This is evaluated on a case study. This draft article is short, interesting, clear and concise. I therefore recommend that it be accepted for publication in AMT with a few minor corrections.

**Response:** Thank you for your thoughtful review. We have incorporated the suggested corrections.

**Comment 1**: Figure 1: I recommend describing the meaning of the red (mean? median?), black and grey (quantiles and/or multiples of standard deviation?) colours.

**Response 1:** As suggested by the reviewer, I have added descriptions to Figure 1 to clarify the meaning of the colors used.

Comment 2: Line 71: the first PHIDP peak appears from Figure 1 to be at 5° rather than 355°.

**Response 2:** As suggested by the reviewer, Page 3 line 76 in the revised manuscript has been revised to correct the first PHIDP peak from 355° to 5°.

**Comment 3:** Line 76: the reference to figure 2a may not be relevant as it does not directly represent ZDR (even though ZDR is derived from Zh and Zv), I would remove it from the text here.

**Response 3:** As suggested by the reviewer, the reference to figure 2a has been removed from Page 3 line 81.

**Comment 4:** Line 79: To make it easier to understand and find the angles, and to make the link with the text, I recommend drawing vertical lines on Figure 1 corresponding to the azimuth angles chosen for Figures 2a, 2b, 2c and 2d.

**Response 4:** As suggested by the reviewer, vertical lines have been added to Figure 1 to correspond with the azimuth angles chosen for Figures 2a, 2b, 2c, and 2d.

**Comment 5:** Line 89: What do the authors think about the values observed at ranges close to 0 km? In figure 2c, we see "biased" Zh values that are lower only at very short distances, and "unbiased" Zv values that are much higher only at short distances in figure 2d.

**Response 5:** Thank you for your observations regarding the radar reflectivity values at close ranges. This anomaly is likely due to near-field effects, interference, or clutter. In future studies, we will investigate this phenomenon further to determine if corrections or calibrations are necessary for short-range observations.

**Comment 6:** Consistency between Figures 2d and 3a: why is there no anomaly at 34° elevation angle in Figure 3a whereas this angle was chosen in Figure 2d to represent an angle for which ZDR has a bias?

**Response 6:** The figure B below presents a zoomed-in snapshot of Figure 3a, specifically focusing on the elevation angles around 27° and 60°. Our objective was to investigate the

source of bias in ZDR (Figure 2d) by selecting elevation angles based on the primary criterion that the biased and unbiased angles must observe the same type of precipitation. As evidenced in the figure below, the 34° angle marks a transition where we start observing the onset of radome-induced bias. The 34° angle was chosen because it is the closest elevation angle to 27° where the anomaly becomes visible, while still observing the same precipitation pattern as at 27°, which remains unaffected by radome-induced bias. The following statement has been appended in Page 3 lines 88 to 90 of the revised manuscript for clarification "For example, the 34° angle marks the transition where the onset of radome-induced bias becomes visible and was chosen as it is the closest elevation angle to 27° (unbiased) that still observes the same precipitation pattern."

---

## Author Response (AR3)

**Response to Reviewers' Comments**

**Reviewer Comment :**

Line 113 and Figure 3. The author mentioned the radar gate or radar bin. Please give the radar gate size (range resolution) in the manuscript.

**Response:**

Thank you for the suggestion. As suggested by the reviewer, details regarding radar gate size (range resolution) has been appended on page 1 lines 36-37 as "The radar performed Plan Position Indicator (PPI) scans with a range gate resolution of 125m and Range Height Indicator (RHI) scans with a range resolution of 25m."